# Research on Fracture Mechanism and Stability of Slope with Tensile Cracks

**Yulin Lu** [1,2,*], **Xiaoran Chen** [3] **and Li Wang** [1,2]

1   School of Civil Engineering, Institute of Disaster Prevention, Beijing 101601, China
2   Key Laboratory of Building Collapse Mechanism and Disaster Prevention, China Earthquake Administration, Beijing 101601, China
3   School of Geological Engineering, Institute of Disaster Prevention, Beijing 101601, China
*   Correspondence: yllu@cidp.edu.cn

**Abstract:** Tensile cracks at the crest of slope will attenuate the stability of slope. The aim of this paper is to investigate the computation of safety factors acting on a clay slope when the slip surface consists of tensile crack and shear surface. Based on the theory of limit equilibrium, an analytical solution for safety factors containing three types of failure mechanisms is presented. The optimal crack depth was obtained by using the principle of minimum safety factor. In the solution, effects of parameters such as crack depth, slope angle, height, cohesion, and internal friction angle on slope stability were discussed. By comparing with the results of previous studies, the rationality of the proposed approach was verified. Results show that consideration of tensile cracks lead to a significant reduction in slope stability, and the safety factor decreases by about 10% compared with the slope without cracks. The law of safety factor varying with crack depth indicates that it first decreases as the crack depth is increased and then increases as the crack depth is further increased. Through the parametric analysis, it is found that the safety factor increases with an increase in cohesion and internal friction angle but decreases with the slope angle and height increase. It is important to note that the optimal crack depth does not exceed one-third of the slope height. Moreover, a highway landslide that occurred in the road running across the Yunnan and Tibet Province was investigated to verify the practicality of the present method.

**Keywords:** slope stability; tensile crack; optimal crack depth; analytical calculation; safety factor

## 1. Introduction

In traditional models of instability of slope, the slip surface of slope is considered as a pure shear surface, irrespective of its form. In practice, slope subjected to crack is a common problem, and the damage is more serious when a landslide occurs. Under the seismic action and groundwater action, the crack will expand rapidly and cause instability [1,2]. The pure shear failure assumed in classical computed model may not be suitable for the actual failure surface of slope. Because tensile cracks are frequently observed at the trailing edges of landslides, the existence of the cracks will reduce stability of slope [1]. However, due to the limitation of pure shear failure model, the effects of tensile cracks on stability of slope were not considered. Therefore, the research on fracture mechanism and stability of slope with tensile cracks is a very valuable work in geotechnical engineering.

In general, slope static stability analysis has matured, but due to the complexity of research on the mechanism of tensile cracks, stability analysis of slopes with cracks should be explored in much more detail systematically. A series of studies has shown that the failure surface of slope is a tensile-shear coupling failure surface, and the results of stability have also demonstrated the rationality of the analysis method [3–8]. However, in the process of slope stability analysis, for complex slip surface, how to determine the ultimate depth of tensile crack has become a prominent issue. Terzaghi [9] determined the expression of critical depth of tensile crack in the vertical clay slope based on the

Rankine earth pressure theory and considered that the depth of tension cracks is not more than half of the slope height. This viewpoint is echoed by some researchers, such as Cousins, Baker, and Zhang et al., who believed that existence of cracks is already dangerously for the stability of slope [10–12]. Chowdhury et al. [13] proposed a simple formula estimating the depth of tensile cracks at the slope tailing edge with arbitrary inclination, but the depth is independent of slope shape. Compared with previous studies, Baker [14] considered the influence of slope geometry on the depth of tensile crack by searching for the most dangerous pure shear slip surface. Utili [15] employed the limit analysis method to investigate the stability of slope with open cracks, based on three types of conditions regarding the presences of cracks, and the same conclusion was also obtained—that the existence of cracks could result in a considerable reduction in stability factors. Michalowski [16] also studied the stability assessment of slopes with cracks which contains two types of cracks, including an open crack that existed prior to slope failure and a formation crack that formed contemporaneously with slope failure by using the limit analysis approach, similarly demonstrating the adverse effects of the different types of cracks. Gao et al. [17] proposed that the mechanisms of cracks formation can be considered part of the internal dissipation process in the limit analysis method, and the stability number obtained by using the limit analysis method was consistent with the conclusion of Baker. Moreover, the results also indicated that neglecting the effect of cracks formation could lead to an erroneous judgement in the stability of slope. For the sake of the accuracy of slope stability evaluation, the influence of tensile crack must be considered. Hence, the mechanisms of cracks and stability analysis has gradually become a research hotspot in slope fields in the future, especially combined with the method of determined cracks value, which is also exactly what this study discusses.

So far, the analysis methods of slope stability mainly include limit equilibrium method, limit analysis method, and numerical simulation method [1,2,9]. Among them, the major advantages of limit equilibrium method are simplicity of concept and convenience of calculation, so that it is widely used all over the world. For the past several decades, several researchers have implemented the limit equilibrium method to analyze the stability of slope [18–20]. After that, Spencer, Baker, Kaniraj and Abdullah, Baker and Leshchinsky, Norly et al. also developed this traditional method to investigate the effects of tensile cracks on the stability of slope [11,21–24]. Unfortunately, the crack depth was only considered as a constant, which was not consistent with actual landslides. On the other hand, in recent years, under the framework of limit analysis method, associating pre-existing cracks of slope stability was also analyzed by different scholars, such as Li et al., Zhou et al., Hou et al., Zhang et al., Zhang et al., He et al., who have been developing 2D and 3D calculated models and analyzing seismic effects, surcharge, and water pressure on the slope with tensile cracks [25–30]. In the above research works, a prominent problem was the crack depth, and location in the slope was still assumed. Therefore, the accuracy of these methods was limited. In fact, crack depth is uncertain before the slope analysis. Although some scholars estimated the depth of tension cracks with corresponding methods, and achieved gratifying results, most of the research about the crack depth had a lack of support of practice verification and in-depth study. The problem of what is a more reasonable crack depth, whether in theory or in practice, has not reached a unified consensus. Therefore, the critical problem to be solved in this study is to give a reasonable value of crack depth.

This study aims to determine the optimal crack depth and analyze stability of slope with cracks by using the limit equilibrium approach. In this paper, in the classical limit equilibrium framework, a static model was firstly proposed to consider the influences of tensile cracks at a slope crest on the stability of clay slope, and the crack depth was defined as a variable. Then, tensile crack was introduced into the slope failure modes with three types of failure surfaces, i.e., toe circle slope, base circle slope, and face circle slope, which are considered in the analysis. Finally, the paper determines the optimal crack depth and minimum safety factor and verifies the rationality of the proposed method by two cases and discusses various geotechnical parameters effects on the crack depth and stability of

slope. In addition, an engineering example was introduced to test the practicability of the present method.

## 2. Formation Mechanism and Value of Vertical Crack

### 2.1. Cracking Mechanism

It is widely known that under the action of external forces, the slope will gradually lose stability, and the original mechanical balance will be destroyed, resulting in a new coordinated deformation. The shear outlet of the clay slope is generally located at the toe of slope. Due to the concentration of shear stress, a large plastic deformation occurs, and its deformation degree is greater than the top of slope. Because of the cumulative effect of deformation, a certain nonuniform settlement will occur near the slope top, thus forming a crack region. In fact, the soil at the top of slope does not have significant tensile strength and thus cannot withstand tension. Therefore, the soil at the top of the slope can usually be simplified as a material that cannot bear the tensile stress.

Rankine earth pressure theory is a calculation method of earth pressure based on the limit equilibrium conditions of soil by studying the stress state in elastic half space [9]. For total stresses, the Rankine active earth pressure beneath a horizontal ground surface, the stress field, is only generated by the weight of soil in infinite half-space, as shown in Figure 1. The vertical stress at any position can be expressed as $\sigma_z = \gamma h$, and it is the maximum principal stress $\sigma_1$. Similarly, the horizontal stress can be expressed as $\sigma_x$, and it is the minimum principal stress $\sigma_3$. Based on the condition of static equilibrium, mathematically,

$$\begin{cases} \sigma_1 = \sigma_z = \gamma z \\ \sigma_3 = \sigma_x = \sigma_1 \tan^2\left(45° - \frac{\varphi}{2}\right) - 2c\tan\left(45° - \frac{\varphi}{2}\right) \end{cases} \tag{1}$$

where $\sigma_1$ and $\sigma_3$ are the maximum and minimum principal stresses, respectively; $\sigma_z$ and $\sigma_x$ are the vertical and horizontal stresses, respectively; $\gamma$ is soil weight, $z$ is the distance to the surface of ground, $\varphi$ is internal friction angle of soil.

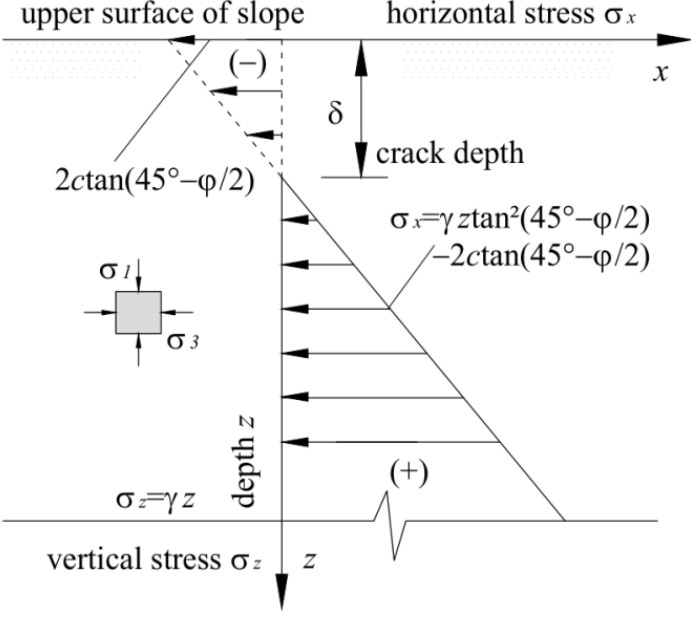

**Figure 1.** Stress distribution of soil.

Because the tensile stress of top soil is very small, it can be considered that $\sigma_3 = 0$. Thus, the crack depth can be expressed as,

$$\delta = \frac{2c}{\gamma\sqrt{K_a}} \tag{2}$$

where, $\delta$ is crack depth, $c$ is cohesion of soil, $K_a = \tan\left(45° - \frac{\varphi}{2}\right)$.

## 2.2. Maximum Value of Tensile Crack

The formation of tensile cracks in slope is extremely complicated, tension is only one of the possible causes of cracks, since there is experimental evidence about cracks caused and deepened by other process, such as cycles of drying and wetting, desiccation, and weathering [15]. Thus, the methods for determining the depth and location of tensile cracks still need further study. A number of researchers have studied the maximum value of crack to evaluate the stability of slope. Terzaghi [9] derived a formula of crack depth of vertical slope, and it can be expressed as,

$$\delta = \frac{1.34c}{\gamma\sqrt{K_a}} \tag{3}$$

Cousins [10] also believed that the crack depth should not exceed half of the slope height. Michalowski [16] considered the crack formation as a part of the failure mechanism by calculating the internal energy dissipation along the crack, and proposed that in the absence of pore–water pressure, the crack depth can be expressed as,

$$\delta = \frac{3.83c}{\gamma\sqrt{K_a}} \tag{4}$$

Chen [31] discussed the effect of Poisson's ratio $\mu$ on soil, and proposed that the crack depth can be expressed as,

$$\delta = \frac{2c\sqrt{K_a}}{\gamma\left[K_a - \left(\frac{\mu}{1-\mu}\right)\right]} \tag{5}$$

These crack formulas are obtained by different methods. However, determining the crack depth remains a challenge to analyze the stability of slope. Up to now, there has been no report about the unified standard of the crack depth or position. Most of the researchers considered the preexisting cracks; that is, the depth of crack is assumed. In general, when a crack is introduced, the crack should not extend beyond the depth of tension. If the crack depth is overestimated, compressive forces will be eliminated, and the safety factor will be overestimated [32–36]. Michalowski proposed that using the static theorem of limit analysis, one can easily prove that the depth calculated from Equation (2) is a rigorous lower bound on the maximum depth of a thin crack in a half-space built of the material with strength governed by the classical Mohr–Coulomb yield condition [16]. However, the crack depth in Equation (4) is the upper bound on the true maximum depth of a stable crack, and a vertical crack with a depth larger than that value will be unstable.

Since there is no uniform standard for equation of the tensile crack, a dimensionless crack coefficient $\lambda$ was introduced to better evaluate the influence of crack value on slope stability in this study. Based on results researched by Michalowski, the upper limit $\lambda$ is set to 2 [16]. It is given by

$$\delta' = \lambda\delta \tag{6}$$

where $\delta$ is crack depth, $\delta'$ is calculated crack depth, and $\lambda$ is the crack depth coefficient and its value range is $0 < \lambda < 2$.

## 2.3. Failure Mode of Slope with Crack

In order to understand the influences of tensile cracks on stability of slope, three types of slope failure were developed in this study. In the stability analysis of clay slope, the slip surface of slope is composed of tension crack and shear surface, and the shear surface is assumed to be circular. According to the location of shear surface outlet, the failure surface can be divided into three types: toe circle slope, base circle slope, and face circle slope, as shown in Figure 2. The location of the center of the circular arc is $O$ (*a*, *b*), *a* and *b* are the coordinates of center of circle, and *r* is radius. When $r = \sqrt{a^2 + b^2}$, it is defined as toe

circle slope; when $r > \sqrt{a^2 + b^2}$, it is defined as base circle slope; when $r < \sqrt{a^2 + b^2}$, it is defined as face circle slope, $h$ is the distance from shear outlet to ground [37,38].

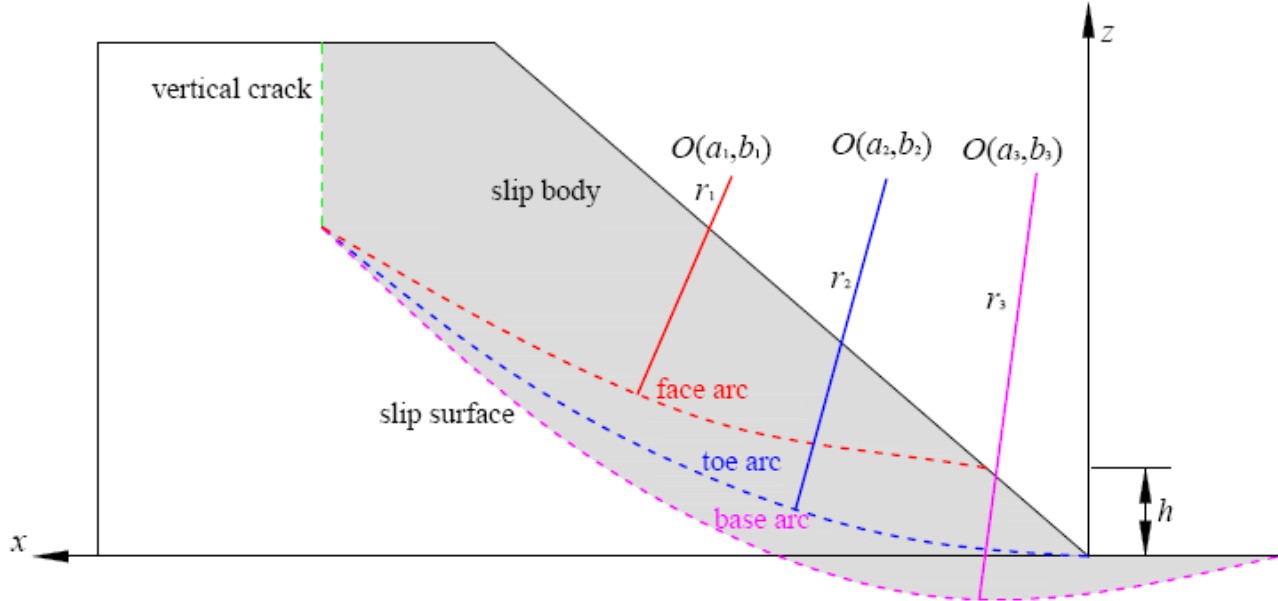

**Figure 2.** Slope failure mode.

Thus, the main assumptions of this research are as follows:

(1) There exists no shear strength for soils in the crack area.

(2) The soil materials follow Mohr–Coulomb failure criterion and relevant flow rule, and the tensile strength of soils is neglected.

(3) The slip body slides along the circular surface with center $O$ ($a$, $b$), and there appears a vertical crack at the slope crest, which forms a combination of the slip surface and the shear slip surface.

(4) The soil in the crack area is simplified as surcharge, and acts on the top of the circular slip surface.

(5) The optimal crack depth is determined according to the principle of minimum safety factor.

## 3. Analytical Method for Safety Factor of Slope with Crack

*3.1. Analytical Method for Safety Factor of Toe Circle Slope*

3.1.1. Calculated Model of Toe Circle Slope

In view of fact that the slip surface of slope is composed of tension crack and shear surface, below the vertical crack area, the slip surface is simplified as a circle surface, and the slip body is treated as a rigid body, as shown in Figure 3. The calculated model of toe circle slope failure is established in the Cartesian coordinate system, and the center of the circle is at the toe of slope. The height of slope is $H$, the angle of slope is $\beta$, the unit weight of soil is $\gamma$, and the vertical crack depth is $\delta$. The parameters of the slip surface are as follows: the coordinates of the center of the circle is $O$ ($a$, $b$), $a$ and $b$ are the coordinate values of $X$ and $Z$ in the direction of the coordinate system shown in Figure 3, the radius of the circle arc is $r$, the length of the slip surface is $l$, and the width of the top of the slip body is $d$. The soil in the crack area is simplified as surcharge, and the surcharge function is defined as $q(x)$.

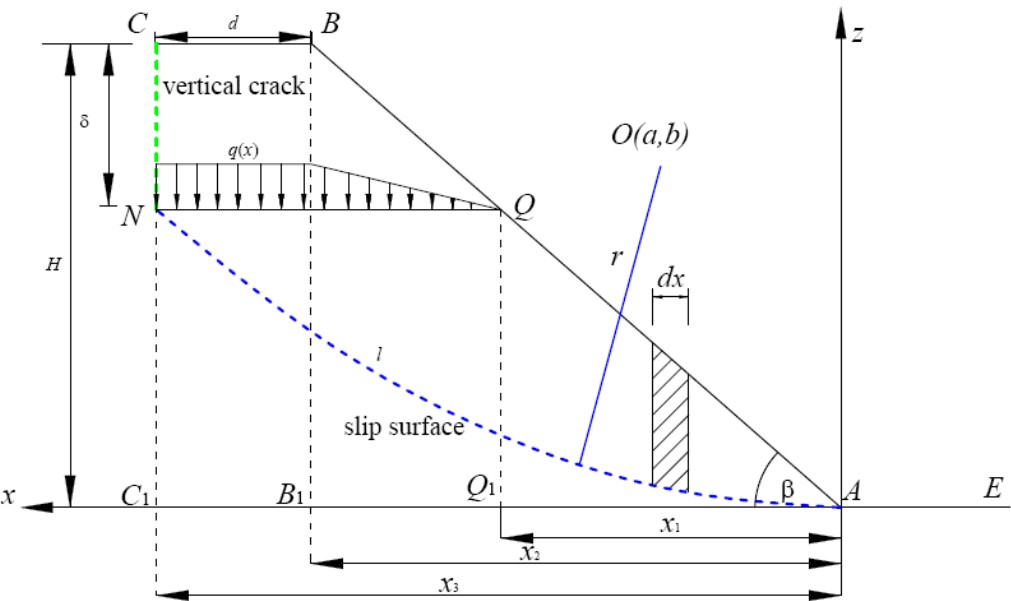

**Figure 3.** Toe circle slope calculated model.

In the calculated model, based on the theory of the Swedish slice method, the slip surface is divided into n slices, and the inter slice forces are plotted in Figure 4 [18]. The weight of any soil slice is $dw$, the length of soil slice is $dx$, the height of soil slice is $h(z)$, the length of the slip surface for soil slice is $dl$, the angle of the slip surface with the horizontal axis is $\theta$, and the normal and shear forces on the bottom of the soil slice are $dN$ and $dT$, respectively. The slip moment of soil slice is $dM$, and the surcharge is $q(x)$.

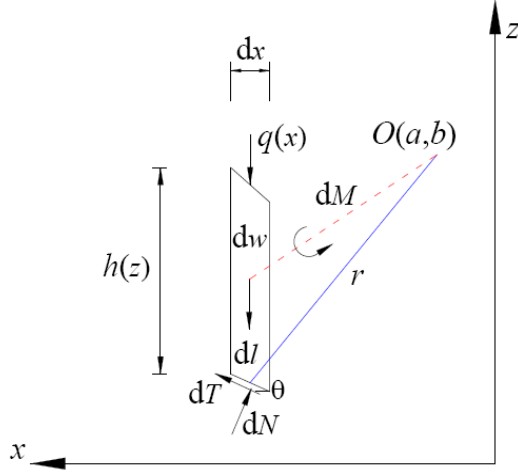

**Figure 4.** Forces acting on the slices subjected to surcharge when the slice forces are neglected.

In the toe circle calculated model, slope face $z_1$, slope upper surface $z_2$, and slip surface $z_3$ can be respectively expressed as follows:

$$\begin{cases} z_1 = x \tan \beta \\ z_2 = H - \delta' \\ z_3 = b - \sqrt{r^2 - (x-a)^2} \end{cases} \tag{7}$$

where weight of the soil slice is expressed as

$$dw = \gamma h(z)dx + q(x)dx \tag{8}$$

The normal and shear forces on the bottom of the soil slice are expressed as,

$$dT = dw \sin\theta \tag{9}$$

$$dN = dw \cos\theta \tag{10}$$

According to the load distribution, the calculation region $x_1$, $x_2$, and $x_3$ can be respectively written as follows:

$$\begin{cases} x_1 = (H - \delta')\cot\beta \\ x_2 = H\cot\beta \\ x_3 = a + \sqrt{r^2 - [(H - \delta') - b]^2} \end{cases} \tag{11}$$

For any soil slice, the relationship between the length of the slip surface $dl$ and the inclination of the slope $\theta$ are expressed as,

$$\begin{cases} \sin\theta = \frac{x-a}{r} \\ \cos\theta = \frac{\sqrt{r^2 - (x-a)^2}}{r} \\ dl = \frac{r}{\sqrt{r^2 - (x-a)^2}} dx \end{cases} \tag{12}$$

The height of soil slice $h(z)$ is written as

$$h(z) = \begin{cases} z_1 - z_3 & 0 \le x \le x_1 \\ z_2 - z_3 & x_1 \le x \le x_3 \end{cases} \tag{13}$$

The surcharge of crack region $q(x)$ is written as

$$q(x) = \begin{cases} 0 & 0 \le x \le x_1 \\ \frac{\gamma}{\cot\beta}(x - x_1) & x_1 \le x \le x_2 \\ \gamma\delta' & x_2 \le x \le x_3 \end{cases} \tag{14}$$

### 3.1.2. Determination of the Safety Factor

In this study, the safety factor was determined by using the limit equilibrium method. The safety factor of slope can be defined as the ratio of resisting moment and sliding moment on slip surface. The soil slice element of slope toe circle is shown in Figure 4, so the safety factor is given by the following expression:

$$F = \frac{M_R}{M_S} = \frac{\int r dT'}{\int r dT} = \frac{\int dT'}{\int dT} = \frac{T'}{T} \tag{15}$$

where $F$ is safety factor, $M_R$ is resisting moment, $M_S$ is sliding moment, $T$ is sliding force, and $T'$ is resisting force.

(1) Calculation of the sliding force

In the toe circle slope, the total sliding force induced by the self-weight, $T_{toe}$, is the result from the shear forces on the bottom of the soil slice, $dT$ shown in Equation (9), acting alone the failure surface $AN$, as

$$\begin{aligned} T_{toe} &= \int_{AN} dT \\ &= \int_0^{x_1} \gamma(z_1 - z_3)\sin\theta dx + \int_{x_1}^{x_3} \gamma(z_2 - z_3)\sin\theta dx + \int_{x_1}^{x_2} \frac{\gamma}{\cot\beta}(x - x_1)\sin\theta d + \int_{x_2}^{x_3} \gamma\delta'\sin\theta dx \\ &= \frac{\gamma(H-\delta')^2}{r}\left[\frac{(a\cot\beta + b)}{2} - \frac{(H-\delta')(1 + \cot^2\beta)}{6}\right] \\ &\quad + \frac{\gamma\cot\beta}{r}\left\{\frac{\cot\beta[H^3 - (H-\delta')^3]}{3} - \frac{[a + (H-\delta')\cot\beta][H^2 - (H-\delta')^2]}{2} + a\delta(H - \delta')\right\} \end{aligned} \tag{16}$$

(2) Calculation of resisting force

The resisting force can be calculated in two parts. One is induced by the cohesion of soil $T'_{toe-c}$ and the other is induced by the friction angle of soil $T'_{toe-\varphi}$.

The $T'_{toe-c}$ acting on the slip surface $AN$ can be written as follows:

$$
\begin{aligned}
T'_{toe-c} &= \int_{AN} c\,dl = \int_0^{x_3} c\frac{r}{\sqrt{r^2-(x-a)^2}}dx \\
&= cr\left(\arcsin\frac{\sqrt{r^2-[(H-\delta')-b]^2}}{r} + \arcsin\frac{a}{r}\right)
\end{aligned}
\tag{17}
$$

Similarly, the $T'_{toe-\varphi}$ acting on the slip surface $AN$ can be written as follows:

$$
\begin{aligned}
&T'_{toe-\varphi} = \tan\varphi \int_{AN} dN \\
&= \tan\varphi\int_0^{x_1}\gamma(z_1-z_3)\cos\theta dx + \tan\varphi\int_{x_1}^{x_3}\gamma(z_2-z_3)\cos\theta dx + \tan\varphi\int_{x_1}^{x_2}\frac{\gamma}{\cot\beta}(x-x_1)\cos\theta dx + \tan\varphi\int_{x_2}^{x_3}\gamma\delta'\cos\theta dx \\
&= \frac{\gamma r\tan\varphi}{2}\left\{\begin{array}{c}[a\tan\beta-(H-\delta')]\arcsin\left[\frac{(H-\delta')\cot\beta-a}{r}\right]\\ -[b-(H-\delta')]\arcsin\frac{\sqrt{r^2-[(H-\delta')-b]^2}}{r}\\ -(b-a\tan\beta)\arcsin\frac{a}{r}\end{array}\right\} + \frac{\gamma\tan\varphi}{6r}\left|\begin{array}{c}\left\{4r^2-[b-(H-\delta')]^2\right\}\sqrt{r^2-[(H-\delta')-b]^2}\\ +2b^3\tan\beta+3a^2b\tan\beta+4a^3+3ab^2-\\ \tan\beta\left\{2r^2+[(H-\delta')\cot\beta-a]^2\right\}\\ \times\sqrt{r^2-[(H-\delta')\cot\beta-a]^2}\end{array}\right| \\
&+ \frac{\gamma\delta'\tan\varphi}{2r}\left\{\begin{array}{c}\sqrt{r^2-[(H-\delta')-b]^2}[(H-\delta')-b]\\ -(H\cot\beta-a)\sqrt{r^2-(H\cot\beta-a)^2}\end{array}\right\} + \frac{\gamma\delta'\tan\varphi}{2}\left\{\arcsin\frac{\sqrt{r^2-[(H-\delta')-b]^2}}{r}-\arcsin\frac{H\cot\beta-a}{r}\right\} \\
&+ \left[\frac{\gamma a\tan\varphi}{2r\cot\beta}-\frac{\gamma(H-\delta')\tan\varphi}{2r}\right]\left\{(H\cot\beta-a)\sqrt{r^2-(H\cot\beta-a)^2}-[(H-\delta')\cot\beta-a]\sqrt{r^2-[(H-\delta')\cot\beta-a]^2}\right\} \\
&+ \left[\frac{\gamma ar\tan\varphi}{2\cot\beta}-\frac{\gamma(H-\delta')r\tan\varphi}{2}\right]\left\{\arcsin\frac{H\cot\beta-a}{r}-\arcsin\frac{(H-\delta')\cot\beta-a}{r}\right\} \\
&- \frac{\gamma a\tan\varphi}{3r\cot\beta}\left|\sqrt{\left[r^2-(H\cot\beta-a)^2\right]^3}-\sqrt{\left\{r^2-[(H-\delta')\cot\beta-a]^2\right\}^3}\right|
\end{aligned}
\tag{18}
$$

The resisting force $T'_{toe}$ can be expressed as

$$
T'_{toe} = T'_{toe-c} + T'_{toe-\varphi}
\tag{19}
$$

Therefore, the safety factor of toe circle slope is given by the following expression:

$$
F = \frac{T'_{toe}}{T_{toe}} = \frac{T'_{toe-c} + T'_{toe-\varphi}}{T_{toe}}
\tag{20}
$$

### 3.2. Analytical Method for Safety Factor of Base Circle Slope

#### 3.2.1. Calculated Model of Base Circle Slope

Compared with the toe circle slope, the base circle slope is characterized by the fact that the shear outlet of the slip surface is located outside the slope. Thus, the calculated model of bottom circle slope can be seen in Figure 5. The point $E$ is the outlet of the slip surface.

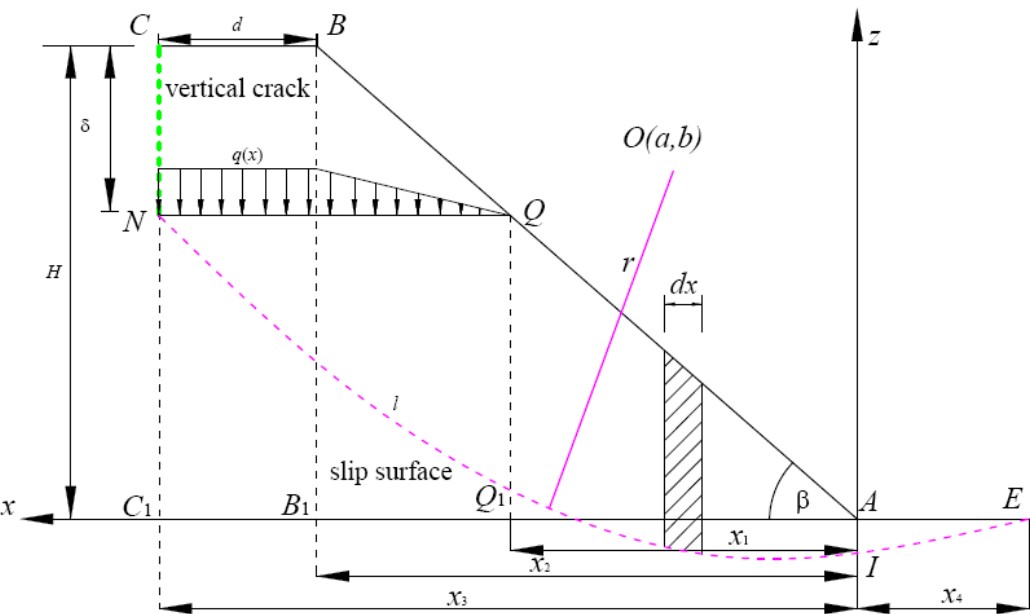

**Figure 5.** Base circle slope calculated model.

In the base circle calculated model, the mathematical formula of slope face $z_1$, slope upper surface $z_2$, and slip surface $z_3$ are the same as the toe circle slope. Similarly, the coordinates of horizontal axis $x_1$, $x_2$, $x_3$, and $x_4$ can be respectively written as follows:

$$\begin{cases} x_1 = (H - \delta') \cot \beta \\ x_2 = H \cot \beta \\ x_3 = a + \sqrt{r^2 - [(H - \delta') - b]^2} \\ x_4 = a - \sqrt{r^2 - b^2} \end{cases} \tag{21}$$

Thus, the height of soil slice $h(z)$ is written as

$$h(z) = \begin{cases} 0 - z_3 & x_4 \le x \le 0 \\ z_1 - z_3 & 0 \le x \le x_1 \\ z_2 - z_3 & x_1 \le x \le x_3 \end{cases} \tag{22}$$

It can be seen from the base circle calculated model that it needs to increase the calculation of resisting and sliding force below the horizontal ground surface compared with the toe circle slope. The slip surface can be represented by *EIN*.

### 3.2.2. Determination of the Safety Factor

The safety factor is also calculated by the limit equilibrium method, as in Equation (15).

(1) Calculation of the sliding force

The sliding force of base circle slope needs to increase the calculated region of *AEI*, and the other region is consistent with the toe circle slope. The sliding force of *AEI*, $T_{base\text{-}AEI}$ can be written as follows:

$$\begin{aligned} T_{base\text{-}AEI} &= \int_{s-EI} \sin \theta \, dw = \int_{x_4}^{0} \gamma \frac{x-a}{r} \left\{ 0 - \left[ b - \sqrt{r^2 - (x-a)^2} \right] \right\} dx \\ &= \frac{\gamma}{r} \left[ \frac{b(r^2 - b^2 - a^2)}{2} + \frac{(b^2)^{\frac{3}{2}} - (r^2 - a^2)^{\frac{3}{2}}}{3} \right] \end{aligned} \tag{23}$$

Thus, the total sliding force of base circle slope $T_{base}$ is expressed as

$$T_{base} = T_{toe} + T_{base\text{-}AEI} \tag{24}$$

(2) Calculation of resisting force

The resisting force of *AEI* is also calculated in two parts, including the resistance $T'_{base-AEI-c}$ caused by cohesion and the resistance $T'_{base-AEI-\varphi}$ caused by friction angle.

$T'_{base-AEI-c}$ acting on the slip surface *EI* can be written as follows:

$$T'_{base-AEI-c} = \int_{EI} cds = \int_{x_4}^{0} \frac{r}{\sqrt{r^2-(x-a)^2}} dx$$
$$= cr\left(\arcsin\frac{\sqrt{r^2-b^2}}{r} - \arcsin\frac{a}{r}\right) \tag{25}$$

Similarly, the $T'_{base-AEI-\varphi}$ acting on the slip surface *EI* can be written as follows:

$$T'_{base-AEI-\varphi} = \tan\varphi \int_{EI} dN = \int_{x_4}^{0} \tan\varphi dw \cos\theta$$
$$= \frac{\gamma \tan\varphi}{r}\left\{ \begin{array}{c} \frac{\sqrt{r^2-b^2}(4r^2-b^2)}{6} + \frac{ab}{2}\sqrt{r^2-a^2}+ \\ \frac{br^2}{2}\left[\arcsin\left(\frac{a}{r}\right) - \arcsin\left(\frac{\sqrt{r^2-b^2}}{r}\right)\right] \\ -ar^2 + \frac{a^3}{3} \end{array} \right\} \tag{26}$$

Thus, the total resisting force $T'_{base}$ of base circle slope is expressed as

$$T'_{base} = T'_{toe} + T'_{base-AEI-c} + T'_{base-AEI-\varphi} \tag{27}$$

Therefore, the safety factor of base circle slope is given by the following expression:

$$F = \frac{T'_{base}}{T_{base}} = \frac{T'_{toe} + T'_{base-AEI-c} + T'_{base-AEI-\varphi}}{T_{toe} + T_{base}} \tag{28}$$

### 3.3. Analytical Method for Safety Factor of Face Circle Slope

Compared with the toe circle slope, the characteristic of the face circle slope, which is the shear outlet of the slip surface, is located on the slope face. Thus, the calculated model of face circle slope can be seen in Figure 6. The calculated model still uses the toe circle slope model, but the slope height needs to be modified to *H-h*. The calculated parameters and methods are consistent with the toe circle slope, which will not be repeated here.

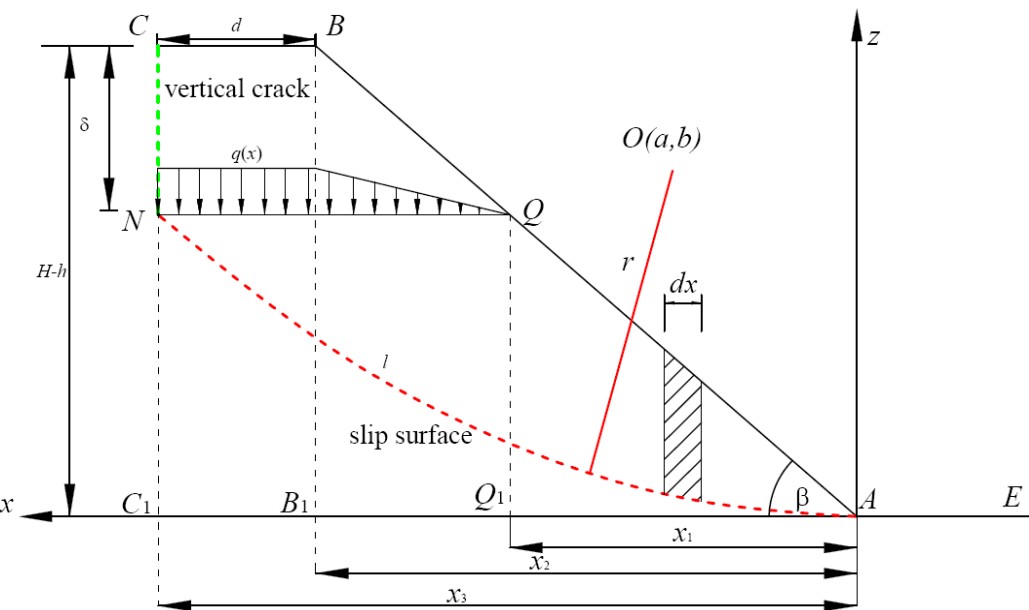

**Figure 6.** Face circle slope calculated model.

### 3.4. Analytical Calculate Process

According to the limit equilibrium method, the safety factors under different conditions can be obtained by explicit methods. In the three failure modes of slope, the vertical crack is considered to be part of the slip surface. Cracks of any depth and location are considered in the optimization process. Thus, the dimensionless coefficient $\lambda$ is not a constant value. To obtain the optimal solution of crack, a minimum safety factor method is proposed for the present mechanism. Taking the calculation of the safety factor as an example, a self-programmed computer procedure by FORTRAN95 languages was used for calculations, and the procedure is expressed in the following steps.

Step1: Input the slope morphology and parameters, including slope height $H$, inclination angle $\beta$, cohesion $c$, and friction angle $\varphi$.

Step2: Input the dimensionless crack depth coefficient $\lambda$ ($0 < \lambda < 2$), which is increased by an unchanging incremental step, and the equal-step is 0.1.

Step3: Determine the slip surface parameters and select the slope failure modes. Then calculate the safety factor, if $|F(i) - F(i + 1)| < 0$, the crack depth optimization process will end. Otherwise, the procedure enters Step2.

Step 4: Determine the minimum safety factor and the crack parameters and evaluate the stability of slope.

The calculation process for the optimal crack depth and the minimum safety factor using the present method are shown in Figure 7.

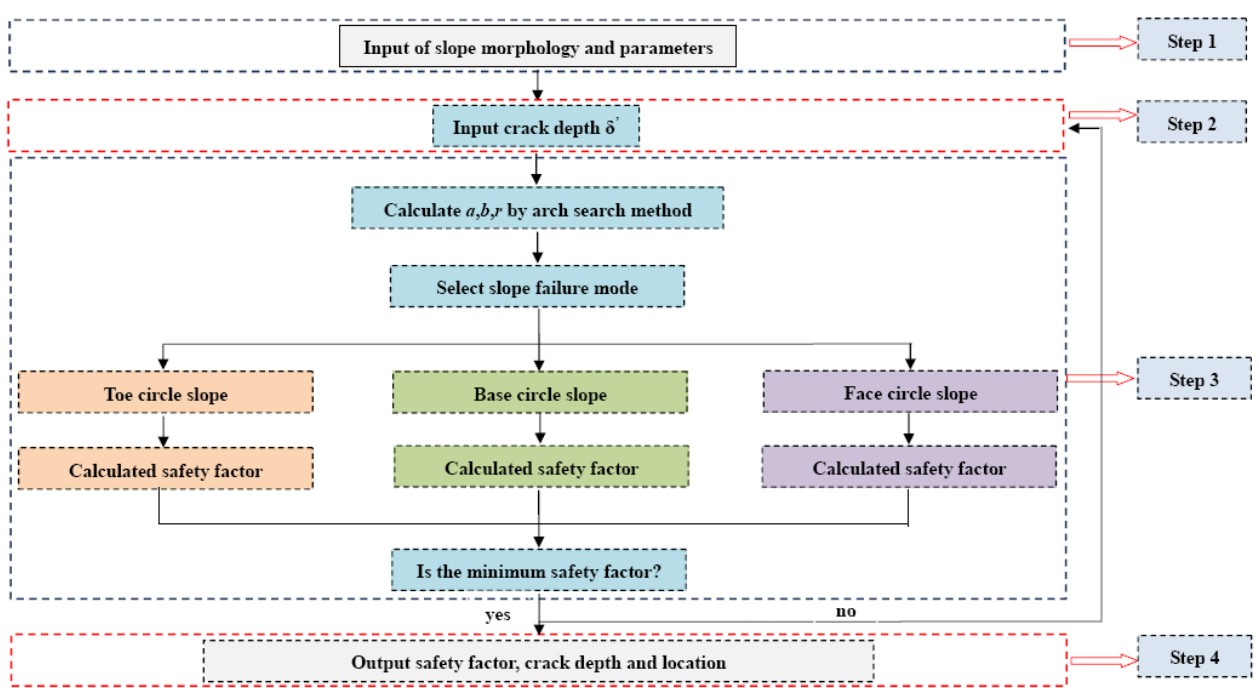

**Figure 7.** Flowchart of the safety factor calculated by the analytical method.

## 4. Results Analysis

### 4.1. Verification of the Present Formulation

In order to verify the rationality of the method proposed in this study, the safety factors of two cases computed by this approach are compared with the values reported in Zhang [39] and Dai [3] studies based on the different methods.

Case 1: The slope height $H = 50$ m, and the physical parameters of the slope are as follows: $c = 58.86$ kPa, $\tan\varphi = 0.2$, $\gamma = 19.62$ kN/m$^3$. In this case, $\lambda = 0$, the vertical crack is set to zero. Table 1 is the result of safety factors computed by the different methods.

**Table 1.** Results of safety factor (case 1).

| Slope Angle $\beta/°$ | $a$/m | $b$/m | Radius $r$/m | *F* Sweden Method | *F* Bishop Method | *F* Present Study |
|---|---|---|---|---|---|---|
| 24 | 34.77 | 110.51 | 117.75 | 1.044 | 1.102 | 1.113 |
| 21.8 | 40.38 | 115.76 | 125.81 | 1.102 | 1.169 | 1.224 |
| 20.0 | 47.74 | 119.38 | 134.84 | 1.158 | 1.254 | 1.304 |
| 18.4 | 53.09 | 128.99 | 143.96 | 1.220 | 1.300 | 1.389 |
| 17.1 | 59.20 | 134.72 | 152.13 | 1.277 | 1.365 | 1.465 |

Case 1 is the failure mode of base circle slope, and the results presented in Table 1 indicate that the present solution is in reasonably good agreement with the case values. It can be seen from Table 1 that the results of the Sweden method are minimal compared with other methods, which is because the influence of inter slice forces on safety factor is not considered. The method proposed in this study is essentially an explicit integration method, and its advantage is to weaken the error caused by ignoring the inter slice forces. The calculated results are close to those of Bishop method, which considered the inter slice force function [18], and the maximum error is about 7.32%, the minimum error is about 0.99%, and the average error is about 4.77%.

Case 2: The slope height $H$ = 20 m, and the physical parameters of the slope are as follows: $c$ = 42 kPa, $\varphi$ = 17°, $\gamma$ = 25 kN/m$^3$, slope angle is 45°, and Poisson's ratio of soil is $\mu$ = 0.3.

The case 2 is the failure mode of toe circle slope. Dai [3] used the strength reduction coefficient method to calculate that the safety factor is $F$ = 1.06, and the crack depth is about 3.1 m, when the parameters of soil is $c$ = 32.793 kPa, $\varphi$ = 17°. The calculation results of present study are crack depth $\delta$ = 3.32 m ($\lambda$ = 1), position of crack $d$ = 10.68 m, slip surface angle $\alpha$ = 28.51°, safety factor $F$ = 1.057.

For the same slope, change the slope angle, and compare the calculation results with the case 2 values in Table 2. It can be seen that the safety factor calculated in this study is basically close to the case values. Although there are differences between calculation methods, the error range of safety factor is generally 7% to 15%, which can meet the engineering requirement [40]. In case 2, the maximum error of safety factor calculated by different methods is about 10.16%, so the error rate is still acceptable. Taking the results of Bishop method as comparison, the results of this study are almost identical. Based on the results of the above two cases, it can be determined that the analytical method in this paper is reasonable.

**Table 2.** Results of safety factor (case 2).

| slope Angle $\beta/°$ | $a$/m | $b$/m | Radius $r$/m | *F* FEM Method | *F* Spencer Method | *F* Bishop Method | *F* Present Study |
|---|---|---|---|---|---|---|---|
| 35 | −4.720 | 72.185 | 72.293 | 1.34 | 1.318 | 1.259 | 1.387 |
| 40 | −3.901 | 66.488 | 66.614 | 1.22 | 1.212 | 1.153 | 1.197 |
| 45 | −9.041 | 75.044 | 75.611 | 1.12 | 1.115 | 1.062 | 1.057 |
| 50 | −3.47 | 66.621 | 66.711 | 1.06 | 1.038 | 0.992 | 0.925 |

### 4.2. The Effects of Parameters on the Stability of Slope

#### 4.2.1. The Effects of Crack Depth and Slope Angle on the Safety Factor

In order to find out the influence of the crack depth on the safety factor, it is necessary to determine the appropriate depth of a crack by performing a series of stability computations

according to the change of the value of crack depth. In case 2, the slope height $H = 20$ m, the slope angle $\beta = 35°{\sim}50°$, the dimensionless coefficient $\lambda \in [0–2]$, and the relationship between crack depth and the safety factor is depicted in Figure 8. The safety factor curve is larger than the critical value 1, the slope is safety. It is clearly found from Figure 8 that the safety factor first decreases as the crack depth is increased and then increases as the crack depth is further increased. The optimal crack depth corresponds to the minimum safety factor, that is, the safety factor of slope with crack is lower than that of without crack. For example, the safety factor for slope angle conditions with $\beta = 35°, 40°, 45°, 50°$, decreases by 6.71%, 7.09%, 7.53%, and 8.16%, respectively, compared with the safety factor of slope without cracks. Figure 8 also shows that the higher slope angle, the more serious the loss of the safety factor and the slope is easier to instability. Additionally, the optimal crack depth is not a constant value, and it increases with the slope angle increasing.

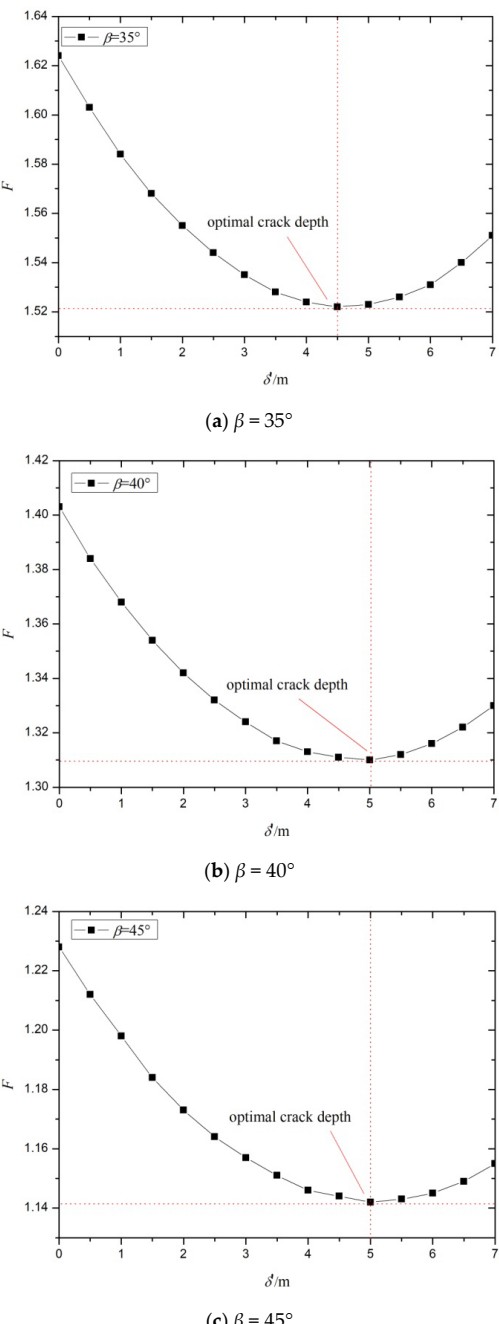

**(a)** $\beta = 35°$

**(b)** $\beta = 40°$

**(c)** $\beta = 45°$

**Figure 8.** *Cont.*

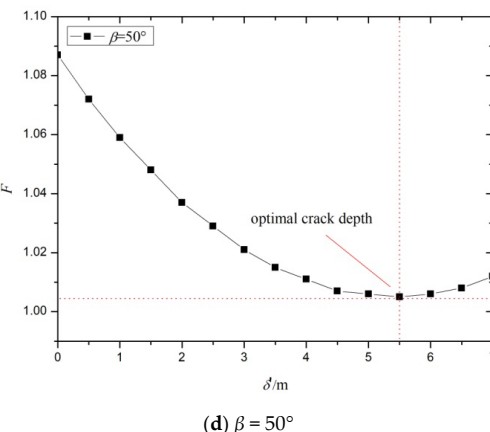

**(d)** $\beta = 50°$

**Figure 8.** Relationship between safety factor and crack depth under different slope angle.

### 4.2.2. The Effects of Crack Depth and Slope Height on the Safety Factor

The range and values of parameters involved in the case 1 and case 2 are the slope height $H = 30\sim90$ m and the dimensionless coefficient $\lambda \in [0\text{–}2]$. It is clearly found from Figure 9 that the safety factor decreases linearly with an increase of crack depth and slope height when the safety factor is less than 1.0 ($F < 1.0$), but the decrease was not obvious. It indicates that the higher slope height, the more likely the slope is to be unstable. Under this kind of condition, the crack depth has only a minor effect on the calculated safety factor. For example, in case 2, for the $H = 90$ m, when the crack depth increases from 0 to 7 m, the value of safety factor decreases only by 1.75% (from 0.858 to 0.843). Similarly, in case 1, for the $H = 90$ m, when the crack depth increases from 0 to 15 m, the value of safety factor decreases by 5.10% (from 0.941 to 0.893).

When the slope safety factor is larger than 1.0 ($F > 1.0$), the law of safety factor varying with the parameter is to reduce at first and increase then with the increasing of crack depth. The above results also indicate that the crack depth has a very obvious effect on the stability of low height slope.

It is important to note that when the slope height is not large, the crack depth does not reach the lower bound on the maximum crack depth given by the limit equilibrium method. Thus, the crack depth should also be limited by different constraints. Terzaghi [9] and Cousins [10] all believed that the crack depth should not exceed half of the slope height. It has been assumed that the slope height of both case 1 and case 2 is 10 m, based on the Equation (2), and the crack depths of case 1 and case 2 are 7.32 m and 5.79 m, respectively. It is obvious that this possibility cannot occur for slope, for similar reasons. Figure 10 explains the relationship between safety factor and crack depth under the low height of slope conditions. It can be concluded that the optimal crack depths of case 1 and case 2 are 3.82 m ($\lambda = 0.52$) and 2.71 m ($\lambda = 0.47$), respectively, which is less than half of crack depths calculated by Equation (2).

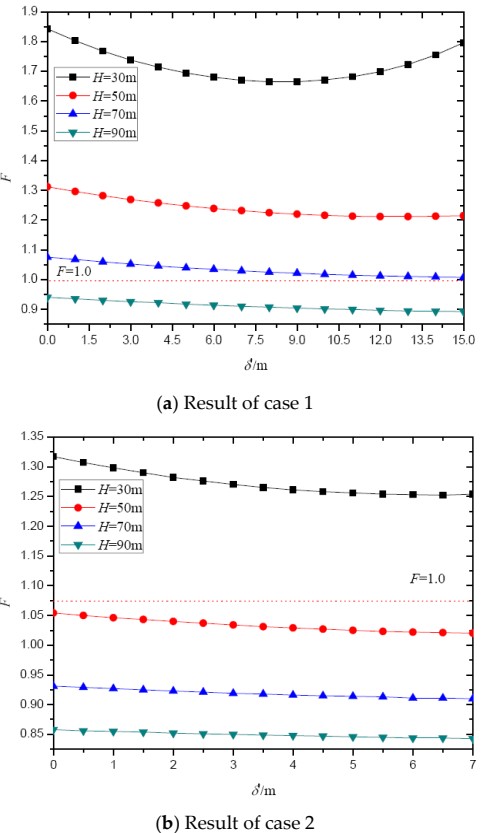

(**a**) Result of case 1

(**b**) Result of case 2

**Figure 9.** Relationship between safety factor and crack depth under different slope height.

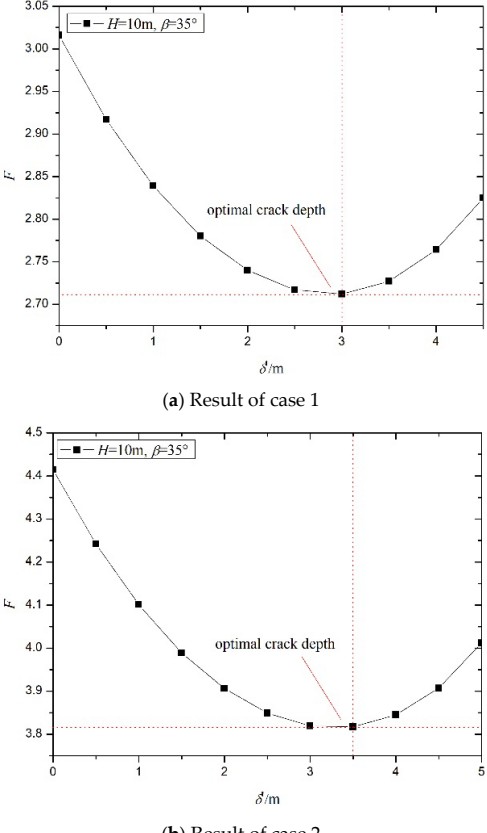

(**a**) Result of case 1

(**b**) Result of case 2

**Figure 10.** Relationship between safety factor and crack depth under low slope height.

### 4.2.3. The Effects of Crack Depth and Internal Friction Angle on the Safety Factor

The safety factor is calculated under the different values of soil internal friction angle $\varphi$ from 10° to 40°, illustrated in Figure 11. As to be expected, the slope is more stable for higher values of the internal friction angle. It is shown in Figure 11 that the higher values of soil internal friction angle, the greater safety factor of the slope. The results of two cases also show that with the increasing internal friction angle, the safety factor gradually increases after a relatively significant decrease, especially on the higher values of friction angle. This indicates that the safety factor is highly sensitive to change in internal friction angle values. Hence, the internal friction angle plays an important role in slope instability subjected to tensile crack. As for an explanation in case 2, when $\varphi$ increases from 10° to 40°, the value of safety factor increases by 110.56% (from 1.421 to 2.992). For the same reason, in case 1, the safety factor also increases by 129.65% (from 1.150 to 2.641). It is noteworthy that the higher values of internal friction angle, the more visible the optimal crack depth effects on the safety factor. This is due to the calculation method of crack depth. When the internal friction angle is very small, the slip surface is mainly composed of the shear surface, so the tensile crack has only a minor effect on the safety factor under this condition. For $\varphi = 10°$, the curves of safety factor of the case 1 and case 2 decrease linearly.

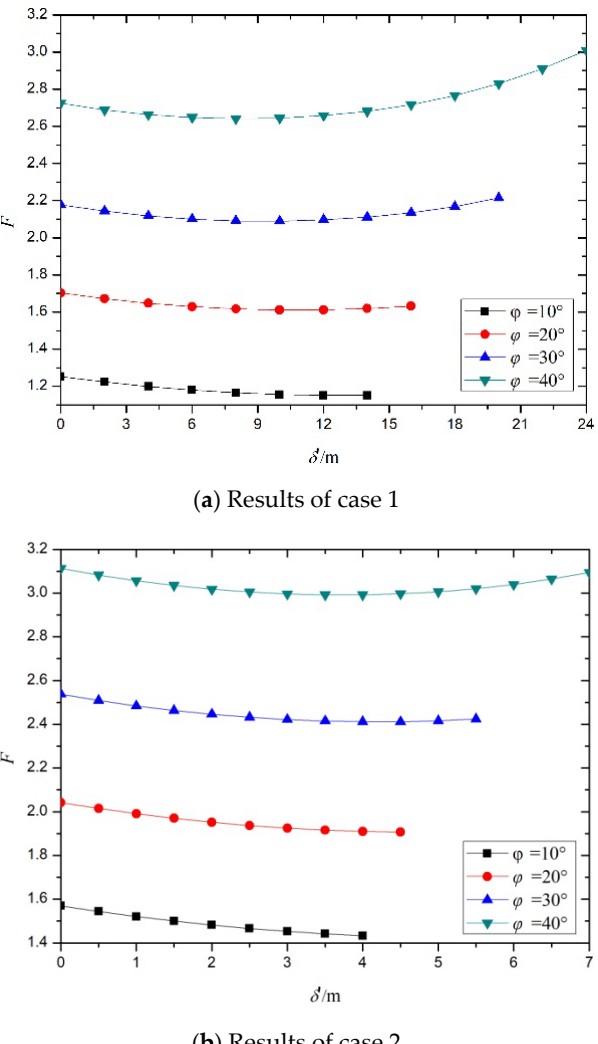

(**a**) Results of case 1

(**b**) Results of case 2

**Figure 11.** Relationship between safety factor and crack depth under different friction angle of soil.

### 4.2.4. The Effects of Crack Depth and Cohesion on the Safety Factor

The crack depth is associated with the $c$, $\varphi$, and $\gamma$. Therefore, for the safety factor and the optimal crack depth, consequently, the cohesion also plays an important role. In this present, the range and values of parameters involved in case 1 and case 2 are: $c = 10$ kPa~70 kPa. It can be seen from Figure 12 that the influences of cohesion on the slope stability of two case results are similar, and the safety factor changes nonlinearly with an increase of cohesion. This variation tendency of the safety factor is also similar to that for the internal friction angle: the safety factor increases for higher values of cohesion. When the safety factor is larger than 1.0 ($F > 1.0$), the safety factor decreases first and then increases with the depth crack increase. It is equally notable that the minimum safety factor corresponding to the optimal crack depth is basically the same. Similarly, when the safety factor is less than 1.0 ($F < 1.0$), the slope is unstable. Quite evidently, it has nothing to do with the crack depth, because the low cohesion can no longer keep the slope stable. Compared with Figure 11, it can be concluded that the effects of the soil physical parameters containing internal friction angle and cohesion on the safety factor are very obvious.

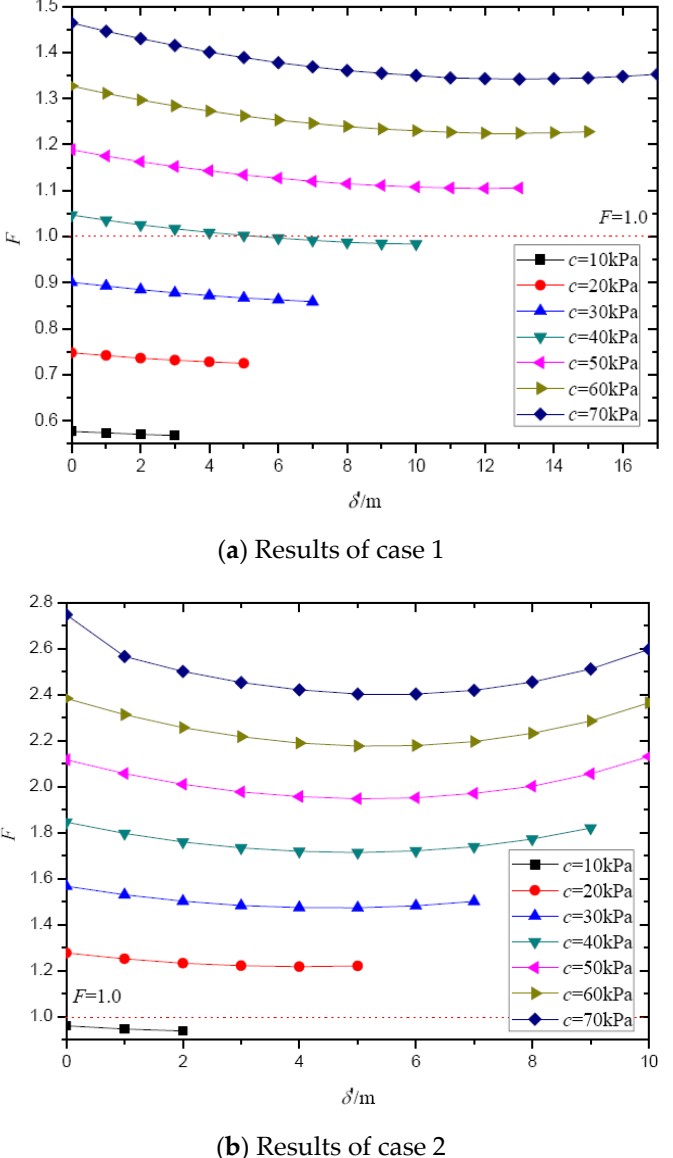

(**a**) Results of case 1

(**b**) Results of case 2

**Figure 12.** Relationship between safety factor and crack depth under different cohesion of soil.

4.2.5. The Effect of Optimal Crack Depth on the Safety Factor

In this study, the optimal crack depth is computed by the principle of minimum safety factor of slope. Therefore, to find the optimal crack depth for any slope, a dimensionless coefficient $\delta'/H$ was introduced to illustrate this. It can be seen in Figure 13a that $\delta'/H$ increases with the increase of cohesion, and the maximum value is $\delta'/H = 0.26$, which means that the optimal crack depth is about one-fourth of the slope height. However, in Figure 13b, $\delta'/H$ decreases with the increase of the internal friction angle, and the maximum value is $\delta'/H = 0.28$. According to the comparison of $\delta'/H$ obtained by the different internal friction angle and cohesion change, the results show that the maximum of $\delta'/H$ from the two methods are approximately equal. For the crack depth, bigger value is not necessarily better, and the optimal crack depth explains this reason. The results of the above two cases show that no matter how the cohesion and internal friction angle change, the optimal crack depth will eventually become consistent. In fact, when choosing the crack depth, this study introduces the dimensionless coefficient $\lambda$ to determine the optimal crack depth. $\lambda$ is not a constant, the value range is $\lambda \in [0–2]$, and its value changes with the change of cohesion and internal friction angle. As pointed out by Terzaghi [9] and Cousins [10], the crack depth should not exceed half of the slope height, and it represents the coefficient $\delta'/H < 0.5$. By contrast, the optimal crack depth in the two cases is far less than half of the slope height, and it can be used as a reference for stability analysis of the slope with tensile cracks.

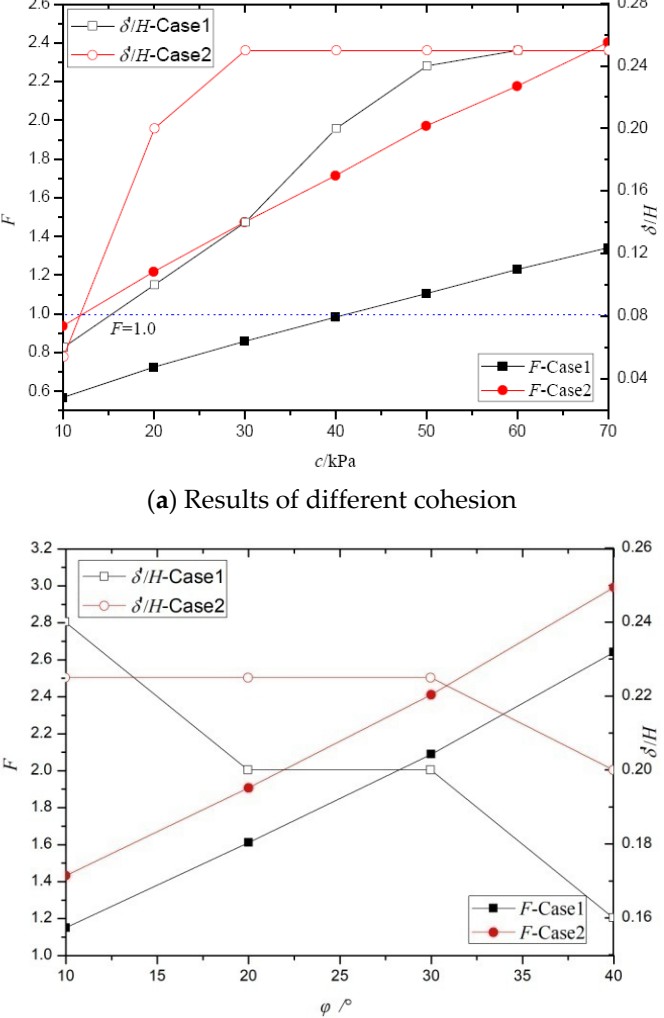

(**a**) Results of different cohesion

(**b**) Results of different internal friction angle

**Figure 13.** Relationship between safety factor and $\delta'/H$.

### 4.2.6. The Effects of Optimal Crack Parameters and Slip Body Morphology on the Safety Factor

If the optimal crack depth $\delta'$ and the horizontal distance of the crack from the slope crest, $d$ (see Figures 3, 5 and 6), is known, the slip body morphology is introduced. When the optimal crack depth is determined, the horizontal location of optimal crack depth can be expressed as

$$
\begin{aligned}
d &= x_3 - x_2 \\
&= a + \sqrt{r^2 - [(H - \delta') - b]^2} - H \cot \beta
\end{aligned}
\tag{29}
$$

Hence, a composite slip surface consists of the tensile crack, and a shear surface is obtained. The slip body morphology of case 2 under different $\delta'/H$ can be seen in Figure 14. For the sake of explanation, the results of slip surface obtained for the values of $\delta'/H$ will be illustrated. It can be seen from Figure 14 that the slip surface moves towards slope face with the increase of values of $\delta'/H$. The $\delta'/H$ ranges from 0 to 0.45, and it shows that the larger its value is, the larger the crack depth is. As is to be expected, for $\delta'/H = 0$, the slip surface is completely shear plane and the safety factor is the maximum. Higher values of $\delta'/H$ can lead to lower safety factors, and the safety factor decreases by approximately 6.23% when the value of $\delta'/H$ varies from 0 to 0.25 (from 1.624 to 1.522). However, the minimum value of safety factor appears when the $\delta'/H = 0.25$, and the crack does not extents to the half height of the slope. It shows that the crack depth is theoretically not unlimited extension, and an optimal value should exist. As shown in the above example, it can be assumed that $\delta'/H = 1$, the slip body morphology coincides with that studied by Utili, the crack coming progressively near to the slope face and at the limit ($d \to 0$) coinciding with it. The $\delta'/H \to 1$, which does not exist in the slope itself. Additionally, it is noted that the crack occurs only at the upper surface of slope, which is an important prerequisite for stability analysis in this study.

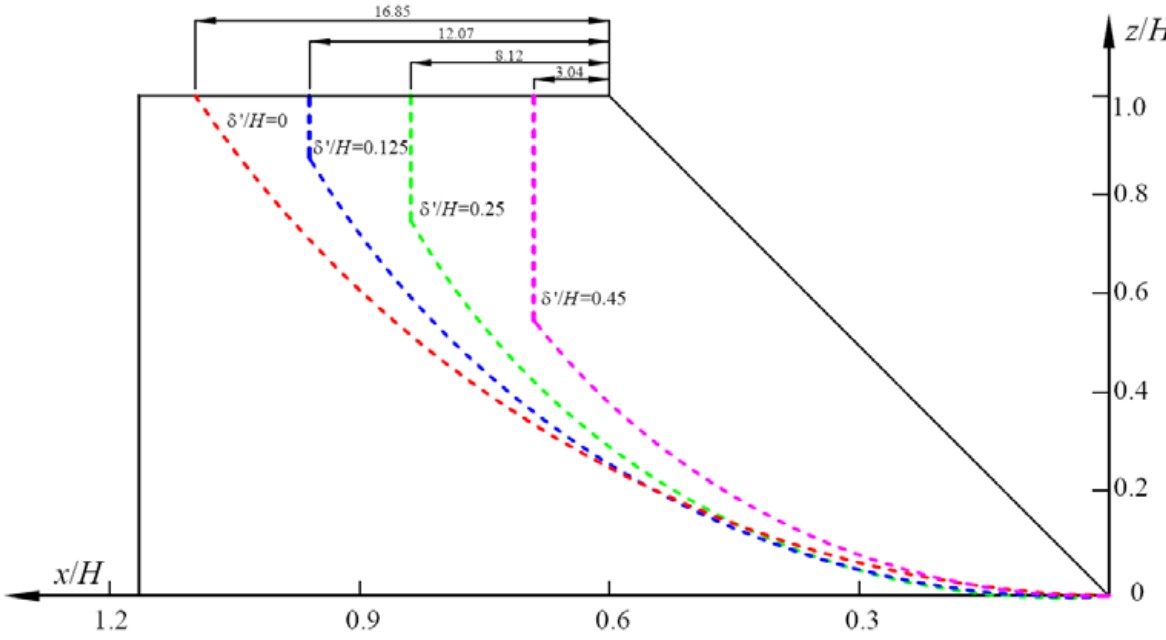

**Figure 14.** Slip body morphology and location of cracks for various values of $\delta'/H$.

## 5. An Engineering Example Application: Determination of the Crack Depth and Safety Factor for a Highway Landslide

Jinchangling-Liuku highway is an important transportation hub in southwest China, and runs across the Yunnan and Tibet Province. A lot of landslides have occurred on both sides of the road, taking a typical one as an example. The landslide parameters are as follows: the height difference between the toe and the top is 103 m; the length of horizontal

projection of the landslide is 120 m; the landslide angle is 50°; the volume of the landslide is estimated to be $5.6 \times 10^4$ m$^3$; the inclination of slip surface is about 40°. The morphology of slope after collapse is shown in Figure 15. The yellow dotted line represents the slide boundary of slip body, and the red dotted line represents the slip surface of landslide. Based upon a great amount of geological investigation data and indoor soil mechanic results, the medium weathered sandstone is the essential component in this area, so this landslide can be treated as a soil-like one. The physical parameters of landslide are as follows: $c$ = 22.4 kPa, $\varphi$ = 28°, $\gamma$ = 19.5 kN/m$^3$.

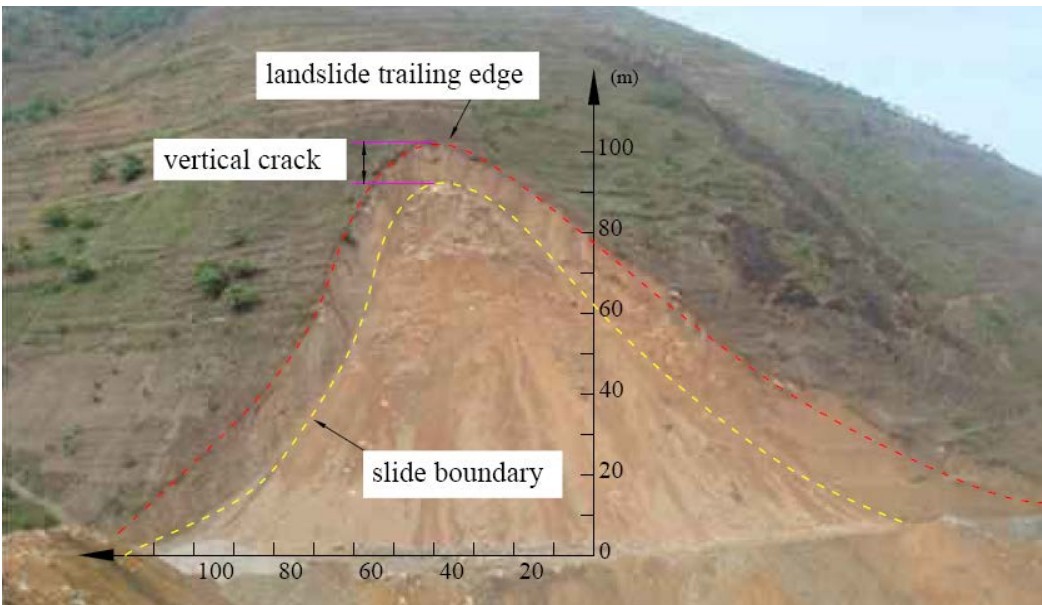

**Figure 15.** Morphology of highway landslide after collapse in Yunnan Province.

According to the engineering geological exploration for the landslide, the morphology of slip body and actual failure surface are shown in Figure 16. Based on the above parameters, the minimum safety factor of the landslide is estimated at 0.98 by using the specification method, and the investigation shows that the crack depth at the tailing edge of landslide is about 8.5m. Since the toe of slope is the outlet of shear slip surface, the toe circle failure model is selected first in this method. Then, through calculation, the optimal crack depth is 11.3m, the horizontal distance from the slope crest is 13.5m, and the coordinate of the center of the circle is $O(-92.61, 201.37)$. Finally, the failure surface can be seen in the purple dotted line in Figure 16, and the corresponding safety factor is 0.923. It is easily found that the results calculated by this study, which contains the slip surface and safety factor, coincidence well with those from the earlier results. The results also show that, under the same condition, compared with the specification method, the safety factor that calculated by the model with tensile crack is slightly smaller. Therefore, it is easily observed that cracks obviously affect the state of landslides, and thus the tensile crack is not neglected in the stability of slope.

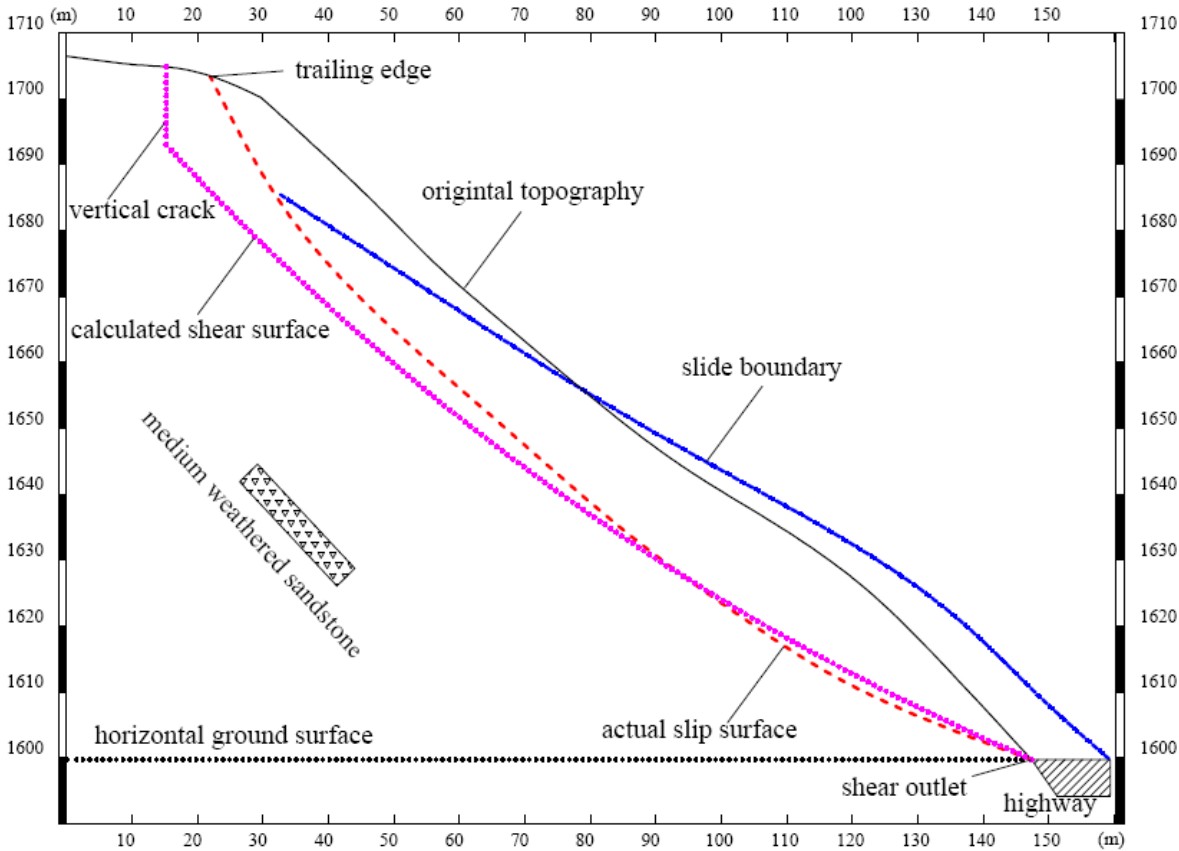

**Figure 16.** Slip surface of highway landslide after collapse.

Moreover, stability of this landslide has been also studied by the finite element method simulation with reduction of the soil strength, and the slip surface can be directly obtained. In numerical simulation, the calculated model is considered as plane strain, with the boundary conditions as follows: the bottom of the model is set as a fully fixed constraint, and both sides of the model are set as horizontal fixed hinge support, and the other sides of the model are freedom. Figure 17 shows the slip surface and the minimum principal stress obtained by finite element analysis program ANSYS. It can be seen from Figure 17b that the minimum principal stress is positive at the trailing edge of landslide, and the other position is negative. In the definition of ANSYS, the tensile stress is positive, and the compressive stress is negative. Thus, the tensile stress at the tailing edge of landslide indicates that the soil will crack, and the vertical height of tensile stress zone is about 14m. The blue dotted line represents crack depth, as given in Figure 17b. Figure 17a is the maximum plastic principal strain of unstable slope, and it shows the shear outlet is located at the toe of slope and run through the top of slope. Based on the distribution of principal strain and principal stress, the slip surface computed by ANSYS is divided into two parts. One is tensile crack and the other is shear surface, as shown in the red dotted line in Figure 17a.

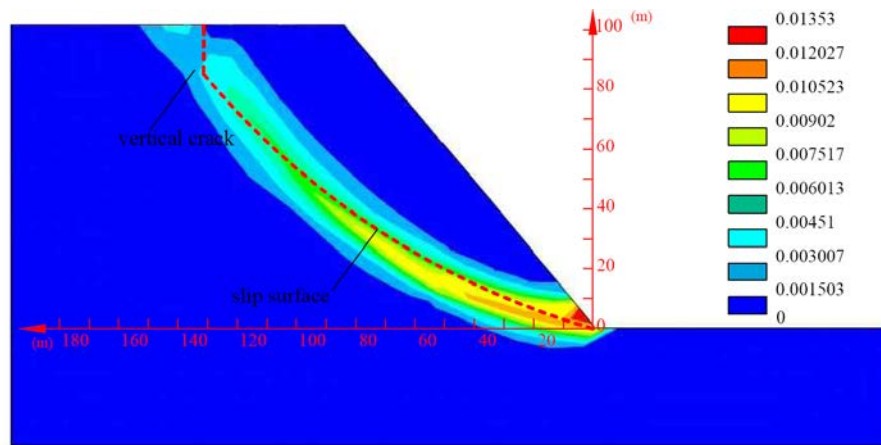

(**a**) Result of the maximum plastic principal strain of unstable slope

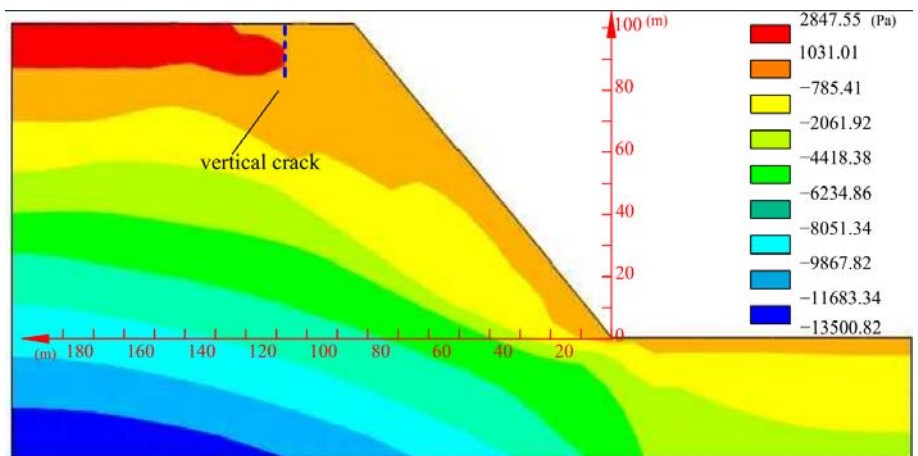

(**b**) Result of the minimum principal stress of unstable slope

**Figure 17.** Results of slope stability calculated by finite element method from ANSYS.

Through contrastive analysis, the slip surface morphology calculated by the finite element method is consistent with the actual landslide. Therefore, the numerical simulation results also verify the rationality of the analytical method proposed in this study.

## 6. Discussion

This paper presents a method for determining the optimal crack depth and safety factor of slope, which depends on the principle of minimum safety factor. However, the tensile failure and shear failure of soil is an interaction relationship, and such interaction or mutual influence has been going on in a synergetic and coupled manner. Due to the limitations of the limit equilibrium method, this coupling has not been considered. As Tang and Kong emphasized, the effects of tension shear coupling on the stability of slope are of critical importance [1,5]. Therefore, the tension shear coupling mechanism should be established in the stability evaluation of the slope with cracks, which will be the subsequent focus of this study.

Crack depth and location has become one of the primary problems and restricts slope stability, but the value of limit crack depth has not been unified, thus the optimal crack depth in this study provides a reference to the stability analysis. In recent years, although many scholars have done a lot of useful work in term of stability evaluation, such as Tang, Utili, Michalowski, and Gao et al., the universality of optimal crack depth also needs further discussion and validation, depending on a large amount of experimental data and measured data [1,15–17]. Therefore, the analytical work will be combined with a series of

model tests to perfect the value of optimal crack depth. This is also inadequate for existing work and needs to be improved in the next step.

The analytical method proposed in this paper depends on strict geometric conditions and boundary conditions, thus leading to the calculated safety factor, which has a certain error compared with the results of other methods. The maximum error is about 10%, which is considered acceptable. The analytical model of slope stability with tensile crack can be also extended and applied to related research, such as seismic slope stability [38].

## 7. Conclusions

The stability of slope with tensile crack is analyzed in this study, and the influence of optimal crack depth and slope parameters on stability are also investigated. The feasibility of the analytical method is verified by comparing with the results of earlier cases, and the applicability of the analytical method is also verified by numerical simulation. To sum up, the main conclusions are obtained as follows:

(1) The safety factor of slope with tensile crack is small compared with the slope without crack, and the value decreases by about 10%. Neglecting the tensile crack will largely overestimate the stability of slope.

(2) The safety factor first decreases as the crack depth increases and then increases as the crack depth further increases. The minimum safety factor, corresponding to the optimal crack depth, is treated as the basis for slope stability evaluation.

(3) The internal friction angle and cohesion of soil play a significant role in the stability of slope subjected to the tensile cracks. The safety factor increases with an increase in cohesion and internal friction angle. As these parameters increase the safety factor will become sensitive, especially when the safety factor is larger than 1.0. Additionally, the safety factor decreases with the increase of slope angle and height, and the larger these parameters are, the more minor the effect of tensile crack on the safety factor is.

(4) The tensile crack depth should not be overestimated, so it is advised to set a reasonable value, that is, the optimal crack depth, to ensure the rationality of slope stability analysis. The results of optimal crack depth show that the maximum, $\delta'/H = 0.28$, does not exceed one-third of the slope height.

**Author Contributions:** Conceptualization, investigation, methodology, Y.L. and X.C.; data processing, L.W.; original draft, Y.L.; writing and editing draft, Y.L. and X.C. All authors have read and agreed to the published version of the manuscript.

**Funding:** Project supported by Hebei Natural Science Foundation (No. E2021512002), Key Laboratory of Building Collapse Mechanism and Disaster Prevention, China Earthquake Administration (No. FZ211104); Langfang Science and Technology Research and Development Plan Project (No. 2021011065).

**Institutional Review Board Statement:** Not applicable.

**Informed Consent Statement:** NOT applicable.

**Data Availability Statement:** The data used to support the finding of this study are available from the corresponding author upon request.

**Acknowledgments:** The authors express appreciation to Haiyan Li for editing and English language assistance.

**Conflicts of Interest:** The authors declare no conflict of interest.

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
