# Peer review of "Research on Fracture Mechanism and Stability of Slope with Tensile Cracks"

_applsci, doi:10.3390/app122412687_

Round 1

Reviewer 1 Report

Dear Authors,

The manuscript presented by you contains interesting content that could enrich the reader-scientist, however, some corrections in existing chapters and extension of the content with chapters not included are required. Below are suggestions for these changes.

ad. 1. Figures 1-6 and nos. 8 and 10, as well as 13-14 are difficult to read. The numbers in the pictures are not clear enough. It should have been enlarged. Figure 15 should also include the town (place of location) in the caption. In general, the clarity of all figures in the manuscript should be reviewed. Enlarge numbers on scales, font thickness, etc.

ad. 2. There is no "Discussion" section in the manuscript, in which the authors of this publication would discuss their thoughts with the publications of other authors. I absolutely require the authors to add this chapter to make this manuscript more meaningful and original.

ad. 3. Authors cited in the manuscript (in the chapter introduction, discussion and reference list) should not be older than 2010, unless 1-2 items are pioneers of the described content of equations, equations, laws, etc.), then there may be among the rest of the new ones. Often, pioneer authors have modernized their ideas and put them in a broader context in their more recent publications. I suggest that the authors of this text of the publication take a close look at their bibliography and make any such modernizations, e.g. literature items 4-9, 13-22, 32.

ad. 4. The number of cited literature in the text should be increased by at least 10 items. it will be possible when the authors add a chapter "discussion". Also, in the introduction chapter, you can cite more literature items and then put them in the list.

ad. 5. Conclusions can also be extended, taking into account the content of the discussion.

Subject to the above conditions, the manuscript may be published.

Good luck

Round 2

Reviewer 1 Report

Dear Authors,

In principle, all my comments have been taken into account and I agree with the authors' answers and corrections. One exception: Please, put the "Discussion" section before the "Conclusions" section. In addition, in the "Discussion" chapter, eliminate the enumeration of 1,2, etc. (because these are not conclusions), but leave the paragraphs. In addition, in each paragraph of "Discussion" put citations of authors. Discussion is a chapter in which the results of one's research are compared with the results of other authors. That is, how the obtained results correspond, or do not correspond, with similar results of other authors. After taking into account this amendment, the article can be published.

Good luck

Reviewer 2 Report

Please refer to the attached file
